# The Usage of Digital Health Mobile-Based Applications among Saudi Population

**DOI:** 10.3390/healthcare11101413

**Published:** 2023-05-12

**Authors:** Naif AlAli, Yasser AlKhudairy, Khalid AlSafadi, Bandar Abduljabbar, Nawfal Aljerian, Abdulrahman M. Albeshry, Najim Z. Alshahrani

**Affiliations:** 1Pediatric Department, AlAflaj General Hospital, Riyadh 16731, Saudi Arabia; 2Anesthesia Department, King Khaled University Hospital, Riyadh 12372, Saudi Arabia; 3College of Medicine, AlMaarefa University, Riyadh 13713, Saudi Arabia; 4Medical Referrals Center, Ministry of Health, Riyadh 11176, Saudi Arabia; 5Department of Emergency Medicine, Faculty of Medicine, King Saud bin Abdulaziz University for Health Specialities, Riyadh 14611, Saudi Arabia; 6Department of Family and Community Medicine, Faculty of Medicine, University of Jeddah, Jeddah 21589, Saudi Arabia

**Keywords:** digital health, mobile, mobile-based applications, medical apps, Saudi Arabia

## Abstract

This study aimed at assessing the extent to which the general Saudi population has embraced digital health medical applications to meet their health-related needs so that the Saudi Ministry of Health and government can appropriately be guided on scaling up digital health across the country. As such, this study was guided by the question of to what extent the Saudi people use digital health mobile-based applications. This was a cross-sectional study utilizing snowballing sampling approach. Frequencies, Chi-square, and Spearman rank correlation statistics were used to offer descriptive and inferential analysis of the variables. The majority of the participants were economically able to afford smart devices that have medical apps, had at least an app on such devices, and highly regarded the benefits of the apps. Unfortunately, their understanding of how to use such apps was limited, and this posed a barrier to embracing digital health alongside difficulty downloading apps and medical ethical concerns. Although there is a willingness, extra effort is needed from the Saudi Ministry of Health and the government to promote the uptake of digital health in Saudi Arabia.

## 1. Introduction

The use of digital information electronically transmitted, stored, and obtained to promote health-related activities and services has been encouraged by the world health organization [1]. As such, novel approaches have been proposed to increase the quality of care and enhance outcomes due to an increased incidence of chronic disorders and difficulties in maintaining adequate screening and follow-up measures [2,3]. Digital health and telemedicine can perform various healthcare procedures [4]. These approaches can significantly enhance the quality of long-distance care, provide patients with suitable preventive and educational options about their disorders, and keep them in touch with their healthcare team in addition to maintaining adequate levels of data privacy [5,6]. Digital health can be used for many purposes, including educational campaigns and research to further enhance the quality of care.

Moreover, properly applying the approaches of digital health and medical applications can also improve access to high-quality and costly approaches that can improve outcomes [7,8]. Mobile and Internet healthcare applications and websites have increased recently [9]. This might be due to increased awareness of the benefits that they offer, including easy monitoring of health [10]. Similar modalities have been reported in developed countries, including online systems and databases that monitor patients with chronic diseases while also offering emergency care [11]. However, the concept of digital healthcare and medical applications within developing countries is still poor [12,13]. For instance, an investigation from Libya estimated that 12% of healthcare workers were not familiar with telemedicine, and only 39% had an adequate understanding of digital healthcare modalities [14].

Attitudes toward the use of digital healthcare and medical applications are markedly affected by individuals’ knowledge and awareness [14]. There are many reasons for this, including the fact that applying these approaches is challenging for individuals in developing countries. These people may have poor training in these modalities and require an introduction to new technologies [15]. Accordingly, adequate assessment of knowledge and awareness is essential to determine the appropriate plan before planning any interventions. However, there is a dearth of evidence in Saudi Arabia exploring the extent to which the general population has embraced medical apps. There was a need to conduct a study to understand the usage of digital health mobile-based applications by the people living in Saudi Arabia. The purpose of this research, therefore, was to explore the usage of digital mobile-health-based applications among Saudis with the aim of suggesting ways in which it can be enhanced.

## 2. Methods

### 2.1. Study Design and Setting

An analytical cross-sectional study was conducted between October and December 2021 in Saudi Arabia.

### 2.2. Study Population and Sampling

All people in Saudi Arabia aged at least 15 years old and having a smart device at the time of data collection were eligible to participate in this study. A snowball approach of sampling was used to identify participants. Any person with a link was urged to send it to a friend(s). Opening the link and filling in the questions in the link implied informed consent to all participants.

### 2.3. Data Collection and Tool

Data were collected using a researcher-designed survey. A link to the survey was shared by the researchers to all their friends, and they were encouraged to share it with as many people as possible across the country. The sharing of the link was through Facebook, Twitter, Instagram, LinkedIn, and WhatsApp. A pre-coded pre-tested questionnaire done specifically for this study contains background questions in addition to other close-ended questions that can investigate awareness and attitudes toward digital health and medical applications. The participants had to answer multiple questions, including demographic factors and questions related to the study objectives. Five different items on the questionnaire measured the knowledge and awareness of the participants. Questions developed regarding knowledge include (1) the number of medical applications that have been downloaded on the mobile device, (2) how much the participant knows about medical applications, (3) the number of official Saudi authorities’ applications that have been used, (4) and whether they faced any difficulties while completing the data in the application. Awareness questions included (1) whether the subjects had heard of digital health, (2) the number of medical applications on the device, (3) how much they know about smart devices’ medical applications, (4) whether they have benefited from those applications in medical uses and situations, (5) and whether they agree that using medical applications on a smart device is necessary.

### 2.4. Data Analysis

Data were entered, coded, and processed using Microsoft Excel and the software Statistical Package for Social Science (SPSS) (Version 23). Descriptive statistics were generated for all variables. A correlation was computed to investigate if there was a statistically significant association between participants’ educational level, number of apps on their devices, and knowledge of smart devices medical apps. All the data were ordinal; thus, Spearman rank correlation statistic was calculated. A Chi-square test was run to determine whether there was a significant difference in using medical apps among participants of differing characteristics. All statistical tests were considered significant at *p* < 0.05.

### 2.5. Ethical Approval

Ethical approval was obtained from Almaarefa University, Riyadh, Saudi Arabia. Confidentiality and anonymity were maintained throughout the study, and the collected data were only used for the purposes described in the study objectives.

## 3. Results

As indicated in Table 1, the majority of the participants were females (67.1%), aged 31 years and above, and Saudi nationals (90.9%). Most of these participants had completed a college level of education (70.5%) and were residing in villa residence type (70.4%) and had owned residence status (71.9%). Most of the participants also had a monthly income of between 6000 and 10,000 (64.9%), did not report having any chronic illnesses (56.5%), were not taking regular medicines (59.1%), and had never faced any medical condition requiring urgent intervention (69.8%).

Table 2 shows the participants’ responses about using medical applications on their smart devices. The majority of the participants had heard of digital health (53.4%) from internet advertisements (24.6%), healthcare workers (15.2%), and family/friends (22.1%). The majority of the participants had at least two medical apps on their smart devices (63.7%). Unfortunately, most (63.5%) of these participants had little knowledge about medical apps on smart devices. Eighty-eight percent (88%) of the participants believed that medical apps positively affect the quality of provided health care and that using medical apps to access personal medical data was appropriate (89.5%). Most (89%) of the participants have never been in a situation of using medical applications to know the medical history to save an injured person. Participants who believed medical data and medical history were not private information were as many as those who believed it was private information.

Table 3 shows that one of the pairs of the variable (number of medical apps on the device and knowledge about smart devices medical applications) was significantly correlated, *r* (507) = 0.178, *p* = 0.000. The direction of the correlation was positive, meaning that highly educated participants tend to have more knowledge about smart devices and medical applications and vice versa.

Table 4 shows that being in a situation of using medical applications to know the medical history for the sake of saving an injured person was statistically significant. The participants who had never been in a situation of using medical applications to know the medical history for the sake of saving an injured person used the medical apps on their smart devices less than their counterparts who had been in such a situation (*X*^2^ = 4.97, *df* = 1, *N* = 736, *p* = 0.026).

Figure 1 shows medical apps that are commonly used in Saudi Arabia to provide healthcare to the people. As shown in Figure 1, the majority of the participants had “Tawakkalna” application on their smart devices (27%), followed by “Sehha” (21%).

As far as the benefits of using medical applications are concerned (Figure 2), most of the participants reported improving the process of making the right decision, followed by providing faster medical intervention and enhancing communication between different healthcare teams.

While the majority of the participants reported having no barrier to the use of medical applications on their smart devices (Figure 3), the three common barriers reported by participants were minimal knowledge about the application, difficulty in downloading the application and using it as well as privacy related to medical data.

## 4. Discussion

As electronic health records gradually replace paper-based methods in healthcare, the manifestation of such changes needs to be assessed in Saudi Arabia. While evidence shows that Saudi Arabia has adopted electronic health records [16,17,18], the usability of such records has not been widely studied. Yet, electronic health records have been found to increase healthcare quality and patient safety by making patient medical information readily available and accessible to any authorized user. While electronic(e) health, digital health, and mobile (m) health can be used interchangeably [18], this manuscript sticks to the use of “digital health in mobile” apps. Digital health is the application of information technology in healthcare by either the health workers through systems or patients through mobile applications [19]. The systems used by health workers in hospitals and clinics largely take on electronic health records that are not the focus of this manuscript, and the mobile applications that create communication between patients and health workers are the focus of this manuscript. Other studies that have studied the usability of medical apps have looked at it through the lens of education/learning apps that have medical information and are not for communication between patients and health workers. For example, Al-ghamdi found increased use of Medscape, Gray’s Anatomy, UpToDate, and Oxford mobile dictionary medical apps in Saudi Arabia relating to where one can find any needed medical information [20]. This study therefore aimed at exploring the extent to which people in Saudi Arabia use medical apps while highlighting the barriers and facilitators to that usability.

In this study, most participants had at least a medical app on their smart devices. In a previous study, Atallah et al. reported similar findings about mentally challenged people [21]; each of them had between one and two applications on their phones. For some reason, people with health concerns in countries where digital health is promoted have been found to have medical apps mostly on their phones [22]. This study finding implies that a reasonable number of Saudis have embraced the government’s call to the utilization of electronic health. In 2018, the Saudi government launched digital health in the country and urged all healthcare facilities and practitioners to embrace the new technology-oriented care [23]. While huge strides have been observed in this regard, the progress warrants the local people to also embrace the move.

The majority of the participants in this study largely believed that the medical apps were good. However, they reported having little knowledge about the use of these medical apps. The knowledge of medical applications was affected by the education level of the participants. The study findings show that the participants who were more educated were more likely to understand medical apps compared to those who were less educated. On the contrary, the majority of the participants were above college-level education, yet, with little knowledge. Implying that other factors could have been responsible for this observation that the study did not explore well.

It is important to note that most of the participants had never suffered from any chronic illnesses, were not taking regular medicines, never faced a medical condition requiring urgent intervention, and had never been in a situation of using medical applications to know the medical history for the sake of saving an injured person. Subsequently, the correlation statistic indicated that the participants who had never been in a situation of using medical applications to know the medical history for the sake of saving an injured person used the medical apps on their smart devices less than their counterparts who had been in such a situation. As such, while it was important to understand digital health considering the general population, it is also important to note that some of the questions asked created a limitation to this study. The respondents had not much knowledge of the asked subject areas. While the majority of the participants had never heard about digital health, their major source of information was the Internet, and less health worker communication. Similarly, other studies have indicated that even among health workers, medical app utilization is low due to personal reasons [24]. Some health workers are still locked up in the traditional life of no smart devices, and even when they obtain smart devices, they are not interested in the functioning of the medical apps on such devices. Given the fact that the Saudi government has intentionally financed the implementation of e-health [25], it should as well widen the health communication to the population (specifically to non-health workers) so that people are aware of the importance and the use of digital health as well. Because evidence also shows that the knowledge of digital health among health workers is lacking [25], wholistic equipping of Saudi people, including health workers on digital health is important and urgently needed. Digital uptake among health workers should be enforced first before scaling it up across the general population.

Digital health education can be emphasized and enforced at the medical/health students’ training level. Or short courses can be developed to be occasionally provided to the health workers in practice. In a different way, Brown and Bewick propose engaging the end users (health workers and people) throughout the process of developing an app or before the formal rolling out of the app [26]. The authors also suggest introducing digital health and electronic health records contained in the curriculum to students so that students can start learning how to support themselves over minor illnesses. This engagement would help minimize the challenges that come with using such apps following their formal release to the public.

This study showed that the participants were financially stable enough to own smart devices that could accommodate medical applications. The majority of the participants were residing in owned apartments and of Villa type, and their monthly income level was between SAR 6000 and SAR 10,000. As such, one would be moved to conclude that they can afford smart devices and the costs involved with the digital health approach.

The importance of digital health records has been demonstrated by many participants. Participants indicated that digital health improves the process of making the right decision, provides a faster medical intervention, and enhances communication between different healthcare teams. Similarly, studies elsewhere have pointed to digital health being of benefit to both patients and healthcare workers [20,27,28]. Al-ghamdi asserts that medical apps on smart devices increase physicians’ accuracy, efficiency, and productivity [20].

For the Saudi Arabia government, this current understanding is good and shows an upward trend toward the desired full penetration of digital health in the country, which was shown to be very low in 2019 [28]. As full medical application penetration is realized, Saudi Arabia will be just like other developed countries that largely rely on digital health implementation for quick interactions between health workers and patients or even in emergency situations where urgent help is needed, and the doctor or physician might not be close. In some instances, patients’ conditions are not so critical to attract physical movement to a health facility. For example, patients with chronic conditions can be supported electronically from their places of residence through their phones.

The following applications have been provided by the Saudi Ministry of Health and can freely be downloaded on any device; Sehhaty, Tawakkalna, Tabaud, Seha, Mawid, and Tataman [29]. Most participants had Tawakkalna and Sehha medical apps on their digital devices. The high use of these two apps is linked to the availability of several services it provides and their mandatory use to obtain services from private and public organizations [22]. Additionally, the high use could be linked to the fact that the Sehha app was the seminal application in 2018 as the Saudi government was launching telemedicine [23]. The Tawakkalna app was later introduced in 2020 as the main contact tracing app for COVID-19-related cases. Contrastingly, in a very recent study, these two apps received the least usability ratings regarding COVID-19 [16]. The respondents reported facing several barriers ranging from increased battery drain and lack of privacy to technical issues [16]. Nevertheless, it is important to observe that the use of medical apps in Saudi Arabia is on the rise and requires slight enforcement.

The commonly reported barriers to digital health were knowledge about application use, the privacy of medical data, and difficulty downloading the apps. The same barriers have been reported previously in Saudi Arabia. Aljohani and Chandra assert that for Saudi Arabia to succeed fully in digital health, the apps should be reliable enough to provide accurate and up-to-date data [28]. Additionally, Aljohani and Chandra put forward social influence as a barrier to digital health; the authors asserted that some health workers and the Saudi population exhibit a culture of boredom pertaining to digital health [28], hence, a need to address it for the success of digital health promotion in the country. An additional barrier that could be worth paying attention to is the language in which most apps have been introduced. While Arabic is largely spoken in Saudi, evidence shows that most of the apps have been introduced in English [30], creating a language barrier for would-be users. It is, therefore, a mandate of the Saudi government to address the barriers above if digital health is to blossom in Saudi Arabia.

The World Health Organization asserts that embracing digital health at both local and distant levels would tremendously address and enhance the struggling healthcare systems in developing countries that have trained doctors, clinicians, equipment, and infrastructure constraints [1]. It is believed that digital health closes the rural–urban divide in many countries [31]. In addition to the recommendations provided throughout this discussion, the following are further provided: (1) A clear policy focus by the Saudi Arabia government is needed to amalgamate the hospital and clinic digital health systems with the way the public ought to fit in. (2) Investment in cross-border health data security in all management systems. This will give confidence to people/patients to freely share their information for the sake of its use in apps. (3) Identifying the best medical apps. Given the fact that some medical apps have received negative reviews from users [20], it is imperative for the Ministry of Health of Saudi Arabia to invest in understanding the best apps and advocate for any required financing for their upscaling.

This study obtains its strength in the snowballing sampling method that was able to reach many people with relevant information and the expert review of the proposal by the institutional review board. Unfortunately, it is limited by the lack of generalization due to the non-randomness of the sampling process and the subjective nature of the questions that would attract responses as understood by the participants. Furthermore, this research does not distinguish between respondents from urban and rural settings. Maybe such a distinction would create different findings.

## 5. Conclusions

This study has shown that digital health among Saudis has several benefits that are on both the side of healthcare and personal health. While many participants already have medical apps on their smart devices, the current study findings indicate that there is a knowledge gap pertaining to their use. Other non-knowledge-related barriers to medical app utilization exist that make people hesitant to both have apps on their smart devices and uncomfortable using them. Given the fact that the Saudi government was at the center of the initiation of digital health in Saudi Arabia, it has an additional role in addressing the barriers hindering the understanding and use of medical apps. This study had several limitations, as indicated throughout the discussion, that make it inappropriate to generalize the results to other study settings. Additionally, further research is needed to fully evaluate the progress on digital health uptake among healthcare workers and how the healthcare workers are motivating the patients or potential patients to embrace it.

## Figures and Tables

**Figure 1 healthcare-11-01413-f001:**
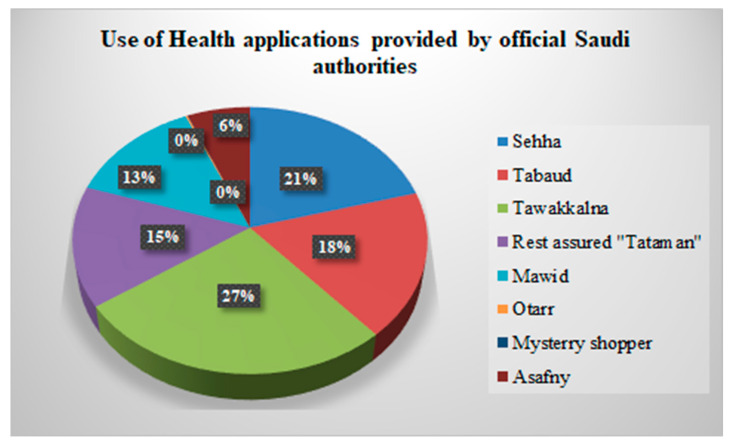
The use of health applications provided by official Saudi authorities.

**Figure 2 healthcare-11-01413-f002:**
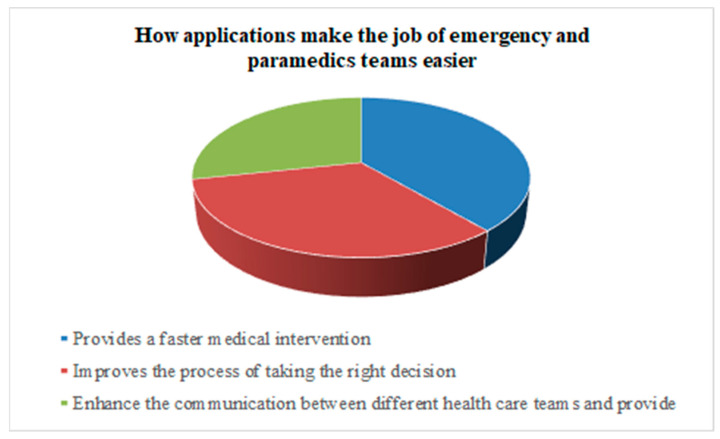
How applications make the job of emergency and paramedics teams easier.

**Figure 3 healthcare-11-01413-f003:**
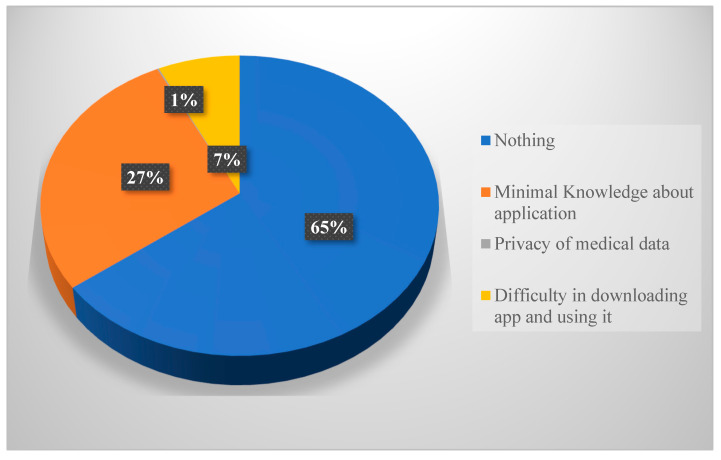
Barriers to use of medical applications on devices.

**Table 1 healthcare-11-01413-t001:** Socio-demographic characteristics of study participants.

Characteristic		Frequency	Percent
Gender			
	Female	494	67.1
	Male	242	32.9
Age			
	15–22 years	110	14.9
	23–30 years	135	18.3
	31–38 years	115	15.6
	39–46 years	138	18.8
	47 or older	238	32.3
Education Level			
	High school	175	23.8
	College	519	70.5
	Elementary	36	4.9
	Primary	6	0.8
Nationality			
	Saudi	669	90.9
	Non-Saudi	67	9.1
Residence type			
	Villa	518	70.4
	Apartment	150	20.4
	A floor of three	68	9.2
Residence Status			
	Owned	529	71.9
	Rented	207	28.1
Monthly Income			
	Less 2000	60	11.3
	2000–5000	127	23.8
	6000–10,000	346	64.9
Suffer from any chronic diseases or allergies			
	Yes	320	43.5
	No	416	56.5
Using regular medications			
	Yes	301	40.9
	No	435	59.1
Faced a medical condition requiring urgent intervention			
	Yes	222	30.2
	No	514	69.8

**Table 2 healthcare-11-01413-t002:** Participants’ responses to questions about medical apps usage.

	Usage of Medical Apps	Count	Percentage (%)
Heard of digital health			
	Yes	393	53.4
	No	343	46.6
Source of information for digital health			
	Internet Advertisement	181	24.6
	Do not know about it	280	38.0
	Healthcare worker	112	15.2
	Family and Friends	163	22.1
Number of medical applications on device			
	One	185	36.3
	Two	141	27.7
	Three or more	183	36.0
Knowledge about smart devices in medical applications			
	Little	467	63.5
	A lot	110	14.9
	Nothing	159	21.6
How applications affect the quality of provided healthcare			
	Positively	648	88.0
	No effect	81	11.0
	Negatively	7	1.0
Using smart devices medical application to access personal medical data is an appropriate way			
	Yes	659	89.5
	No	77	10.5
Completed filling up the required data related to history of illnesses in the applications			
	Yes	247	33.6
	No	489	66.4
Ever been in a situation of using medical applications to know the medical history to save an injured person			
	Yes	81	11
	No	655	89
Medical history in applications useful to any of the medical teams in a previous emergency			
	Yes	48	6.5
	No	130	17.7
	Never been in a situation where it was needed	558	75.8
Some medical applications are part of the operating system of smart devices which does not require downloading.			
	Yes	250	34.0
	No	486	66.0
Medical data and history of illness are private information that you do not prefer to share			
	Yes	368	50.0
	No	368	50.0

**Table 3 healthcare-11-01413-t003:** Intercorrelations, Means, and Standard Deviation for three variables (N = 509).

Variable	1	2	3	M	SD
1. Education Level	---	−0.002	0.057	1.83	0.54
2. Number of medical apps on device	---	---	*0.178* *	2.00	0.85
3. Knowledge about smart devices’ medical applications	---	---	---	1.58	0.82

* Italic and asterisk values indicate statistical significant (i.e., *p* < 0.05).

**Table 4 healthcare-11-01413-t004:** Chi-square statistic to determine difference for use of medical apps among participants of selected characteristics.

		Used Any of the Saudi Authorized Medical Apps			
Variable	*n*	Yes	No	X^2^	*p*
Monthly Income:				1.44	0.488
Less 2000	60	12	48		
2000–5000	127	18	109		
6000–10,000	346	49	297		
Suffers from chronic illnesses or allergies:				0.14	0.708
Yes	320	58	262		
No	416	71	345		
Using regular medication:				0.20	0.658
Yes	301	55	246		
No	435	74	361		
Been in a situation of using medical applications to know the medical history for the sake of saving an injured person:				4.97	*0.026* *
Yes	81	7	74		
No	655	122	533		

* Italic and asterisk values indicate statistically significant (i.e., *p* < 0.05).

## Data Availability

Data used in this analysis can be obtained by contacting the corresponding author.

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
