# Peer review of "The Usage of Digital Health Mobile-Based Applications among Saudi Population"

_healthcare, 2023, doi:10.3390/healthcare11101413_

Round 1

Reviewer 1 Report

1.Introduction section:

(1). P64,P65: However, there is a dearth of evidence in Saudi Arabia exploring the extent to  which the general population has embraced medical apps.

This gives a conclusion that needs to be supplemented by the current state of the relevant research, the differences between your research and others' research.

(2). Add the purpose and meaning of your research in order to justify the need for the research

2.Data collection and tool:

The description of the questionnaire content, listing 1,2,3,4,5 more clearly

3.Results:

The explanation of each table, with a percentage of instructions, with a chart will be more clear

4.Supplementary note: The sampling of interviewees has advantages and disadvantages and limitations. Whether the distribution of interviewees is reasonable.

Author Response

1.Introduction section:

(1). P64,P65: However, there is a dearth of evidence in Saudi Arabia exploring the extent to  which the general population has embraced medical apps. This gives a conclusion that needs to be supplemented by the current state of the relevant research, the differences between your research and others' research

Authors response: further relevant studies have been added to the discussion part to highlight the difference between this research and others’ related research.

(2). Add the purpose and meaning of your research in order to justify the need for the research

Authors response: The purpose of the study has been added in the introduction.

2.Data collection and tool:

The description of the questionnaire content, listing 1,2,3,4,5 more clearly

Authors response: The numbering system has been adopted as suggested.

3.Results:

The explanation of each table, with a percentage of instructions, with a chart will be more clear

Authors response: We tried our best to addressed it per you suggestion.

4.Supplementary note: The sampling of interviewees has advantages and disadvantages and limitations. Whether the distribution of interviewees is reasonable.

Authors response: This has been taken care of under study strengths and limitations.

Reviewer 2 Report

Dear authors,

I think the subject of the study is very interesting. After a careful reading of the manuscript, I have found some considerations that I would like to mention to you:

Introduction

The introduction is well-structured and provides a clear rationale for the study.

Methods

Ethical approval

-          Line 105: Provide the registration number of the scientific committee.

Results

-          In lines 109-110 the authors state: “As indicated in table one, majority of the participants were females (67.1%), aged at least 31 years, …” However, Table 1 provides data showing lower age ranges 15 - 22 years and 23 - 30. In fact, according to the table, there were more females in the latter age range than in the 31 - 38 age range. I'm sorry, but I don’t understand this and need clarification.

-          Line 120: “healthcare workers (15.2%). and family/friends (22.1%).” Please, remove the dot.

-          Lines 133-136: “To investigate if there was a statistically significant association between participants’ educational level, number of apps on their devices, and knowledge of smart devices medical apps, a correlation was computed. All the data was ordinal; thus, Spearman rank correlation statistic was calculated.” I think this paragraph should be part of the material and methods instead of the results section.

-          Lines 143-144: “To determine whether there was a significant difference on using medical apps among participants of differing characteristics, a Chi-square test was run.” Likewise, I think that this paragraph should be part of the material and methods instead of the results section.

Discussion

-          Lines 178-186: This paragraph is more appropriate for the Introduction section than for the Discussion section and, if necessary, should be relocated. On the other hand, the first paragraph of the Discussion section is usually reserved for recalling the main objective of the study with the intention of refocusing the readers' attention.

-          A subsection including the limitations and strengths of the study should be added at the end of the Discussion section.

-          Lines 210-211: “it is also important to note that some of the asked questions created a limitation to this study.” This should be included in that subsection of studio limitations and explained.

-          Lines 211-217 and 242-243:

o   “The respondents had not much knowledge of the asked subject areas. Contrastingly, other studies have indicated that even among health workers, medical app utilization is low due to personal reasons [21]. Some people are still locked up in the traditional life of no smart devices and even when they get smart devices they are not interested in the functioning of the medical apps on such devices. Therefore, digital uptake among health workers needs to be enforced first before scaling it up across the general population.”

o   “However, as this recommendation suffices, evidence also shows that the knowledge of digital health among health workers is lacking.”

The authors discuss the non-use or low use of medical apps by healthcare workers in different paragraphs. It would be more correct to unify all this information in the same paragraph to facilitate the reading of this section.

-          Lines 267-271: In my opinion, this paragraph is more appropriate for the Introduction section.

In general terms, the discussion needs to be restructured to improve its readability and comprehension, adding a section on the limitations and strengths of the study at the end.

Author Response

Dear authors,

I think the subject of the study is very interesting. After a careful reading of the manuscript, I have found some considerations that I would like to mention to you:

Introduction

The introduction is well-structured and provides a clear rationale for the study.

Authors response: Thank you.

Methods

Ethical approval

-          Line 105: Provide the registration number of the scientific committee.

Authors response: This has been inserted.

Results

-          In lines 109-110 the authors state: “As indicated in table one, majority of the participants were females (67.1%), aged at least 31 years, …” However, Table 1 provides data showing lower age ranges 15 - 22 years and 23 - 30. In fact, according to the table, there were more females in the latter age range than in the 31 - 38 age range. I'm sorry, but I don’t understand this and need clarification.

Authors response: Looking at the age categories of the participants, the totals for categories 12-22, and 23-30 were lower than the totals for categories of above 30 years. Meaning majority of the participants were aged at least 31 years (31 years and above). I have, however, re-written this in a simpler language for you to understand.

-          Line 120: “healthcare workers (15.2%). and family/friends (22.1%).” Please, remove the dot.

Authors response: Dot has been removed.

-          Lines 133-136: “To investigate if there was a statistically significant association between participants’ educational level, number of apps on their devices, and knowledge of smart devices medical apps, a correlation was computed. All the data was ordinal; thus, Spearman rank correlation statistic was calculated.” I think this paragraph should be part of the material and methods instead of the results section.

Authors response: This paragraph has been moved to the methods part.

-          Lines 143-144: “To determine whether there was a significant difference on using medical apps among participants of differing characteristics, a Chi-square test was run.” Likewise, I think that this paragraph should be part of the material and methods instead of the results section.

Authors response: This paragraph has been moved to the methods part.

Discussion

-          Lines 178-186: This paragraph is more appropriate for the Introduction section than for the Discussion section and, if necessary, should be relocated. On the other hand, the first paragraph of the Discussion section is usually reserved for recalling the main objective of the study with the intention of refocusing the readers' attention.

Authors response: We perceive that there is enough content in the introduction part yet this very information is good for the introduction particularly to harmonize what would create confusion to the reader. As such, we have maintained it but added a sentence at the end of the first paragraph to recall the aim of the study.

-          A subsection including the limitations and strengths of the study should be added at the end of the Discussion section.

Authors response: Limitations and strengths of the study have been added.

-          Lines 210-211: “it is also important to note that some of the asked questions created a limitation to this study.” This should be included in that subsection of studio limitations and explained.

Authors response: This has been added as suggested.

-          Lines 211-217 and 242-243:

o   “The respondents had not much knowledge of the asked subject areas. Contrastingly, other studies have indicated that even among health workers, medical app utilization is low due to personal reasons [21]. Some people are still locked up in the traditional life of no smart devices and even when they get smart devices they are not interested in the functioning of the medical apps on such devices. Therefore, digital uptake among health workers needs to be enforced first before scaling it up across the general population.”

o   “However, as this recommendation suffices, evidence also shows that the knowledge of digital health among health workers is lacking.”

The authors discuss the non-use or low use of medical apps by healthcare workers in different paragraphs. It would be more correct to unify all this information in the same paragraph to facilitate the reading of this section.

Authors response: The discussion about health workers has been moved to the same paragraph.

-          Lines 267-271: In my opinion, this paragraph is more appropriate for the Introduction section.

Authors response: The mentioned lines throughout the reviewer’s comments do not correspond to the actual lines in the original manuscript. It is, therefore, hard for us to know which exact sentences the reviewer is pointing to in this comment.

In general terms, the discussion needs to be restructured to improve its readability and comprehension, adding a section on the limitations and strengths of the study at the end.

Authors response: There has been restructuring and strengths and limitations to the study have been added.

Reviewer 3 Report

This paper addresses digital health in mobile-based applications.  Unfortunately, I learn little from reading this paper.  First, you are not consistent in talking about mobile health and digital health in general, particularly in the discussion.  I think you should define digital health.  In the discussion, you talk about the EHR but do not tie it to mobile digital health. What apps are available?  What functions do they provide? How do patients learn about apps? How do they download? In most cases, one clicks on an app and then clicks on install.  Should be an easy task?  How are the apps advertised?  What other apps, such as game apps are typically on the devices? Do clinicians promote the apps? To add value and understanding to the paper, you need much more detail about functionality. What is the level of teaching health literacy and how to use the medical apps?

To understand the value of the snowball approach, I need some idea of the population of the regions in your study. What were the 5 items on your questionnaire? In Table 2 for source, there could be more than one source.  Might be important. On line 110, I don't understand the comment "aged at least 31 years." Table 1 has data starting at 15 years.

I have no idea what a Sehha or a Tawakkalna application is.  Describe, as well as the other terms in Figure 1.

I find Figures 2 and 3 not particularly useful as that data had already been provided. Figure 3 seems to show that privacy was 0%, yet in your discussion, you note that privacy concerns were a reason for not using the device.  Are the apps free?

In your discussion, mobile digital health seems to get lost in contrast to EHR. The conclusion is weak.  No discussion on how to improve on use and value of mobile digital health.  What can the health industry do?  What can government do? How can you educate people? How can you prove value?  The topic is important, but you need to increase the information in the article to make it useful.

The language is mostly acceptable except for an occasionally awkward sentence.

Author Response

This paper addresses digital health in mobile-based applications.  Unfortunately, I learn little from reading this paper.  First, you are not consistent in talking about mobile health and digital health in general, particularly in the discussion.  I think you should define digital health.  In the discussion, you talk about the EHR but do not tie it to mobile digital health. What apps are available?  What functions do they provide? How do patients learn about apps? How do they download? In most cases, one clicks on an app and then clicks on install.  Should be an easy task?  How are the apps advertised?  What other apps, such as game apps are typically on the devices? Do clinicians promote the apps? To add value and understanding to the paper, you need much more detail about functionality. What is the level of teaching health literacy and how to use the medical apps?

Authors response: By the scope of this manuscript, we are not able to cover all the questions posed by this reviewer. Rather, this paper focused on making the reader understand the extent to which digital health is in Saudi Arabia, the facilitators, and barriers, as well as the necessary strategies for scaling up digital health.

To understand the value of the snowball approach, I need some idea of the population of the regions in your study. What were the 5 items on your questionnaire? In Table 2 for source, there could be more than one source.  Might be important. On line 110, I don't understand the comment "aged at least 31 years." Table 1 has data starting at 15 years.

Authors response: About age: Looking at the age categories of the participants, the totals for categories 12-22, and 23-30 were lower than the totals for categories of above 30 years. Meaning majority of the participants were aged at least 31 years (31 years and above). I have, however, re-written this in a simpler language for you to understand.

I have no idea what a Sehha or a Tawakkalna application is.  Describe, as well as the other terms in Figure 1.

Authors response: Tawakkalna is the official Covid-19 application in the Kingdom of Saudi Arabia to prevent the spread of the Covid-19 virus and is developed by the Saudi Data and Artificial Intelligence Authority (SDAIA). Seha is E-health saudi application. It was introduced in 2018 to allow individuals to have face-to-face visual medical consultations with their doctors on their smartphones. The app is designed to enable audio–video communication, as users can login into the app, communicate directly with a specialist and have their cases diagnosed through the app. Hence, a specialist answers users' inquiries, conducts the needed medical consultation and provides the required medical procedure.

I find Figures 2 and 3 not particularly useful as that data had already been provided. Figure 3 seems to show that privacy was 0%, yet in your discussion, you note that privacy concerns were a reason for not using the device.  Are the apps free?

Authors response: Figure 2 &3 provide barriers and facilitators to the use of medical apps. These have not been provided elsewhere unless you point out the exact sentences (lines). Figure 3 has been put right.

In your discussion, mobile digital health seems to get lost in contrast to EHR. The conclusion is weak.  No discussion on how to improve on use and value of mobile digital health.  What can the health industry do?  What can government do? How can you educate people? How can you prove value?  The topic is important, but you need to increase the information in the article to make it useful.

Authors response: More information to address these concerns has been added to the discussion part of the manuscript.

Round 2

Reviewer 2 Report

Dear authors,

I would like to express my gratitude for the modifications that have been made to the manuscript. It is my belief that the alterations have resulted in an enhancement of the quality of its presentation.

From my perspective, I do not find any grounds for objecting to its publication in its current state.

Kind regards.

Reviewer 3 Report

For the most part, I think you have done an excellent job in addressing all the reviewers' comments. What should be included is a matter of opinion, and I bow to your choice.  Good job.